# T-Regulatory Cells Confer Increased Myelination and Stem Cell Activity after Stroke-Induced White Matter Injury

**DOI:** 10.3390/jcm8040537

**Published:** 2019-04-19

**Authors:** Sydney Zarriello, Elliot G. Neal, Yuji Kaneko, Cesario V. Borlongan

**Affiliations:** 1Center of Excellence for Aging and Brain Repair, University of South Florida Morsani College of Medicine, Tampa, FL 33612, USA; zarriello@health.usf.edu (S.Z.); elliotneal@health.usf.edu (E.G.N.); ykaneko@health.usf.edu (Y.K.); 2Biomedical Research Concentration, Scholarly Concentrations Program, University of South Florida Morsani College of Medicine, Tampa, FL 33612, USA

**Keywords:** bone marrow-derived stem cells, mesenchymal stem cells, T-regulatory cells, white matter injury, stroke, oligodendrocytes, oligodendrocyte precursor cells, oxygen–glucose deprivation, reoxygenation

## Abstract

Stroke-induced hypoxia causes oligodendrocyte death due to inflammation, lack of oxygen and exacerbation of cell death. Bone marrow-derived stem cells (BMSCs) possess an endogenous population of T-regulatory cells (T_regs_) which reduce secretion of pro-inflammatory cytokines that lead to secondary cell death. Here, we hypothesize that oligodendrocyte progenitor cells (OPCs) cultured with BMSCs containing their native T_reg_ population show greater cell viability, less pro-inflammatory cytokine secretion and greater myelin production after exposure to oxygen-glucose deprivation and reoxygenation (OGD/R) than OPCs cultured without T_regs_. OPCs were cultured and then exposed to OGD/R. BMSCs with or without T_regs_ were added to the co-culture immediately after ischemia. The T_regs_ were depleted by running the BMSCs through a column containing a magnetic substrate. Fibroblast growth factor beta (FGF-β) and interleukin 6 (IL-6) ELISAs determined BMSC activity levels. Immunohistochemistry assessed OPC differentiation. OPCs cultured with BMSCs containing their endogenous T_regs_ showed increased myelin production compared to the BMSCs with depleted T_regs_. IL-6 and FGF-β were increased in the group cultured with T_regs_. Collectively, these results suggest that BMSCs containing T_regs_ are more therapeutically active, and that T_regs_ have beneficial effects on OPCs subjected to ischemia. T_regs_ play an important role in stem cell therapy and can potentially treat white matter injury post-stroke.

## 1. Introduction

Although stroke rates are declining, cerebrovascular accidents present a major obstacle for public health in the United States. Stroke is responsible for approximately 1 in 20 deaths in Americans [1]. Ischemic stroke comprises 87% of all strokes [1]. The loss of blood perfusion to brain tissue creates a hypoxic environment for endothelial cells, neurons and other cells critical for providing support to the neurovascular unit, resulting acutely in necrosis [2,3], and the ensuing secondary cell death plagued with activation of immune cells and pro-inflammatory cytokines, such as tumor necrosis factor alpha (TNF-α), interleukin 1 beta (IL-1β), interleukin 6 (IL-6) and other elements of the innate and adaptive immune systems [4]. Prolonged inflammation can have detrimental effects on brain tissue and can cause further cell death [5].

Clinical safety and efficacy of stem cell transplantation is still being determined [6]. Bone marrow-derived stem cells (BMSCs) have been widely investigated for their potential use in stroke treatment due to their performance in clinical trials, though lack of a sufficient control and sample size hinders their progression into clinical use [7,8]. Endogenous BMSCs and transplanted BMSCs mobilize and home into the inflamed site after an ischemic episode [9]. Self-renewing BMSCs reside in the bone marrow and have the capacity to differentiate into multiple cell lineages, including fibroblasts, adipocytes, placental cells, skeletal myocytes, chondrocytes and others both in vivo and in vitro [10]. In past studies, researchers have extensively used the markers CD271 and CD146 to identify BMSCs [10]. Additional markers of multipotent BMSCs include platelet-derived growth factor alpha and CD105 [10]. BMSCs have long been of interest since the 1800s, when a German scientist hypothesized that fibroblasts originated in the bone marrow [11]. More recently, they have been investigated for the purpose of tissue renewal and their anti-inflammatory secretome, which could be useful in mitigating inflammation caused by pathologic conditions. Mesenchymal stem cells possess an attractive track record of conferring anti-inflammation [12]. When administered in the acute phase of ischemic injury, stem cell transplantation is thought to attenuate oxidative stress, inflammation and cell death [13]. T-regulatory cells (T_regs_), a subtype of T-cell, have recently discovered immunosuppressive properties. T_regs_ express the biomarkers FoxP3, CD25 and CD4 and have been found to exist as a small subpopulation of BMSCs [14]. The native T_reg_ population, at present about 0.4% of BMSCs, was shown to confer increased cell viability via anti-inflammation in a set of neurons subjected to oxygen-glucose deprivation and reoxygenation (OGD/R). After OGD/R, pro-inflammatory cytokine IL-6 decreased with increasing concentrations of T_regs_ in co-culture [14]. There was a loss of immunosuppressive function after deletion of the FoxP3 allele [15]. Interestingly, the number of T_regs_ varied inversely with basic fibroblast growth factor beta (FGF-β), suggesting that increasing the number of T_regs_ beyond that of the native population within BMSCs may have deleterious effects on the survival and differentiation of the BMSCs [14]. M2 microglia possess neuroprotective abilities and have been shown to facilitate white matter repair [16]. T_regs_ can mitigate inflammation after ischemia by influencing macrophage polarization toward the more regenerative M2 phenotype [17,18,19]. Indeed, a decrease in T_regs_ is seen after ischemic stroke, which could play a crucial role in exacerbating an inflammatory state and causing further cell death [20,21,22].

Gray matter has been a focus as a potential therapeutic intervention site for ischemic stroke, but the role of white matter damage following ischemia has been neglected. Stroke in areas of the brain containing white matter tracts represents a large minority of patients [23]. Myelin-producing oligodendrocytes create a sheath around axons that allows saltatory conduction. Diffuse excitotoxic death has been observed in oligodendrocytes after ischemia [24,25]. Decreased white matter integrity after ischemic stroke has been associated with long-term cognitive impairment [26]. Additionally, increasing evidence has implicated tissue plasminogen activators’ potentiating effects on ameliorating white matter injury [27]. Edaravone, a potent radical scavenger, was shown to protect oligodendrocyte precursor cells (OPCs) against oxidative stress [28]. Because stem cells possess similar effects on oxidative stress, this may be an avenue by which stem cell transplantation could provide neuroprotection. Adipose-derived mesenchymal cells (ADMSCs) transplanted into the infarct region after ischemia were shown to induce oligodendrocyte progenitor cell (OPC) proliferation in rats [29]. ADMSC administration was also associated with increased myelination [29]. Induced pluripotent stem cell-derived neural stem cell transplantation also increased white matter viability after stroke in pigs [30]. 

Mechanisms by which stem cells improve recovery after stroke remain unclear, despite many promising studies examining their effects. In the present study, we seek to elucidate the role of the endogenous population of T_regs_ contained within BMSCs in ischemic stroke and white matter recovery. We hypothesized that depletion of T_regs_ before BMSC co-culture with OPCs subjected to OGD/R would result in decreased myelination (Figure 1). Furthermore, we expected increased levels of IL-6 in OPCs treated with T_reg_ depleted BMSCs, which would confirm the anti-inflammatory properties of T_regs_ that have been widely studied. We also predicted that the increase in pro-inflammatory cytokines would affect oligodendrocyte development from OPCs. It is crucial that investigation of the stem cell-brain interaction is pursued, especially the under-explored white matter injury after stroke.

## 2. Materials and Methods

### 2.1. Groups

We used 5 groups to assess the effects of BMSCs and T_regs_ on ischemic OPCs. The first group served as the control and contained OPCs not exposed to oxygen-glucose deprivation (OGD) and not co-cultured with BMSCs (control). The second group contained OPCs exposed to OGD but not co-cultured with BMSCs (OGD). The third group contained OPCs that were exposed to OGD and co-cultured with BMSCs containing T_regs_ (OPCs with BMSCs T_reg^+^_). The fourth group contained OPCs that were exposed to OGD and co-cultured with BMSCs that had been depleted of their T_regs_ (OPCs with BMSCs T_reg^−^_). Lastly, the fifth group contained OPCs that were exposed to OGD and co-cultured with BMSCs that were run through the depletion column, but not depleted of their T_regs_ (OPCs with BMSCs no magnet). 

### 2.2. Cell Culture

Rat OPCs were cultured in vitro as described previously [2]. A total of 40,000 OPCs (ScienCell, Carlsbad, CA, USA) were suspended in 400 μL OPC medium (ScienCell, Carlsbad, CA, USA) and grown in poly-l-lysine-coated glass 8-well plates at 37 °C. The medium contained 500 mL of basal medium, 5 mL of OPC growth supplement and 5 mL of penicillin/streptomycin solution (ScienCell, Carlsbad, CA, USA). Purified BMSCs were obtained and stored in aliquots in dry ice (AllCells, Almeda, CA, USA). After the cells reached confluence, they were subjected to an oxygen-glucose deprivation and reoxygenation (OGD/R) condition (5% CO2, 95% N2) for 90 min. The cells were reoxygenated and co-cultured at 37 °C with BMSCs with or without T_regs_, at a concentration of 40,000 cells per well for three more days (Figure 2). OPCs and BMSCs in co-culture were suspended in the same medium and 8-well poly-l-lysine plates were used for the primary culture of OPCs alone. The co-culture was created by directly inoculating the culture medium with the appropriate number of BMSCs with or without their T_regs_.

### 2.3. T_reg_ Depletion

BMSCs were thawed and run through the depletion column approximately one hour before the co-culture was created. Anti-CD4 and CD25 antibodies were used to label T_regs_. The bound antibodies were then conjugated with magnetic microbeads (Miltenyi Biotec, Bergisch Gladbach, Germany). Magnetically labeled cells were isolated by passing the cell suspension through a column containing magnetic metal substrate. CD25^+^ positive cells were depleted by passing the BMSCs through the column. The resulting eluent was a population of BMSCs that did not contain native T_regs_. To mimic the conditions of the OGD with BMSCs T_reg^−^_ group, the OGD with BMSCs no magnet group was similarly run through the depletion column, but the cells were not magnetically labeled and the magnet was inert. We compared this group to the other BMSC groups in order to determine whether the mechanical stress from the depletion column affected the secretion of cytokines or the expression of myelin basic protein (MBP) or O4. 

### 2.4. Measurement of Extracellular Cytokines: IL-6, FGF-β

After OGD/R, culture medium was centrifuged at 3000× *g* for 15 min and supernatant was collected and frozen at −80 °C. The supernatant was later processed and analyzed using ELISA kits specific to IL-6 (ThermoFisher Scientific, Waltham, MA, USA) and FGF-β (Abcam, Cambridge, MA, USA) with absorbance measured at 450 nm on a Synergy HT plate reader (Bio-Tex Inc., Houston, TX, USA).

### 2.5. Immunocytochemistry

After cell culture was completed, cells were fixed for 30 min in 4% paraformaldehyde, permeabilized with a 0.1% tween solution, and incubated with fluorescent antibodies with affinity for MBP and oligodendrocyte marker O4 at the manufacturer’s recommended dilutions. Comparison of cell viability between the control and OGD groups was performed using a DAPI nuclear stain (Vector Laboratories, Burlingame, CA, USA). Images of approximately 50 cells/picture were captured in 32 randomly selected areas (1 mm^2^, n = 32/group) per treatment condition to quantify cell staining for MBP and/or O4 (ImageJ, National Institutes of Health, Bethesda, MD, USA). Stained cells were digitally captured under microscope (20×). In our 8 well plates, one photo was captured from each quadrant in each well (four photos per well). The cells with positively stained nuclei only for MBP and O4 were not counted as MBP+ and O4+ respectively when calculating the percentage of positive cells, as this does not indicate typical morphology. O4+ and MBP+ cells were counted as positive if cells showed processes extending beyond the cell nucleus, as this is consistent with the shape generally assumed by OPCs and oligodendrocytes. Clusters of cells were not considered O4+ or MBP+ if obvious processes extending beyond their nuclei could not be identified.

### 2.6. Statistics

For categorical independent variables with more than one dependent continuous variable, a one-way ANOVA test was used to determine significance. A *p*-value less than 0.05 was considered significant. All statistical tests were conducted using IBM SPSS Statistics software (IBM, Armonk, NY, USA). The sample size (eight replicates) was based on our previous study allowing rigorous testing of cultured cells for recognizing a 25% statistically significant difference between treatments and controls [31]. Pairwise comparisons were made with the Tukey-HSD post-hoc test and Fisher’s PLSD post-hoc test for the cell count data and ELISA data respectively.

## 3. Results

### 3.1. OGD Resulted in Decreased Cell Viability

Cells were counted using ImageJ software after they were imaged with a confocal microscope. Exposure to OGD/R caused a reduction in number of surviving OPCs to 56.03% of the control (*p* < 0.05). This survival rate was similar to what was obtained in our previous study [14]. The OPCs co-cultured with the BMSC population that was run through the column but not exposed to the magnet did not show a significantly different cell count than the other BMSC groups, confirming that running the cells through the column did not alter cell survival. 

### 3.2. OPCs Show a More Mature Oligodendrocyte Phenotype

Cells exposed to OGD/R and co-cultured with BMSCs showed greater expression of the O4 marker than both the control and the OGD/R group, suggesting greater maturity (*p* < 0.001) (Figure 3). There was no significant difference between BMSC groups. Cells that displayed staining only in the nuclear region were not considered positive for the purposes of the quantitative analysis. Cells that did not have processes extending beyond the nucleus staining for either O4 or MBP were unlikely to be of later oligodendrocyte lineage. 

### 3.3. T_regs_ Confer Increased Myelination after OGD/R

Groups co-cultured with BMSCs showed increased myelination compared to groups not co-cultured with BMSCs (*p* < 0.001), again implying that BMSCs secrete growth factors necessary for oligodendrocyte differentiation (Figure 4). The BMSC T_reg^+^_ and BMSC no magnet groups showed increased MBP positive cells compared to the BMSC T_reg^−^_ group (both ps < 0.01). Since the BMSC no magnet group was not depleted of its T_regs_, this suggests that the presence of the endogenous T_reg_ population within BMSCs influences the outgrowth and production of myelin processes. Qualitatively, the BMSC T_reg^+^_ and BMSC no magnet group had more delineated and robust myelin processes than the BMSC T_reg^−^_ group. No significant differences in staining for either O4 or MBP were seen between the OGD with BMSCs T_reg^+^_ group and the OGD with BMSCs no magnet group, and this therefore suggests that running the cells through the column did not alter O4 or MBP expression. 

### 3.4. T_reg^+^_ Groups Showed Increased Extracellular Cytokines

Extracellular secretion of cytokines FGF-β and IL-6 was different among the BMSC groups (*p* = 0.05 and *p* < 0.0001, respectively). Interestingly, the BMSC T_reg^+^_ showed increased IL-6 production compared to both other groups (both ps < 0.001) (Figure 5). There was increased FGF-β secretion in the BMSC no magnet group compared to the BMSC T_reg^−^_ group (*p* < 0.05). The BMSC T_reg^+^_ group displayed more secretion of FGF-β than the BMSC T_reg^−^_ group, suggesting that the BMSCs were more active in this group (*p* < 0.05). There was no significant difference between the FGF-β secretion in the BMSC T_reg^+^_ group and the BMSC no magnet group (*p* = 0.72). We assayed both the control and OGD supernatants in both ELISAs, but no results were obtainable because both ELISA kits were customized to detect human cytokines only. 

## 4. Discussion

The present study tested the effects of T_reg_ cells within a BMSC population on stroke-induced white matter injury in OPCs. Our main findings are as follows: (1) OGD reduced cell viability of OPCs; (2) OPCs treated with BMSCs containing their endogenous T_reg_ population displayed increased myelin production compared to BMSCs with depleted T_regs_; (3) co-culture with BMSCs with and without T_regs_ induced differentiation of immature OPCs into myelin-producing oligodendrocytes compared to control; and (4) extracellular cytokines were increased in the T_reg^+^_ groups. In addition, treatment with BMSCs in general (with or without T_regs_) was associated with increased OPC differentiation into myelinating oligodendrocytes. This implies that stem cell transplantation into injured white matter may aid in healing and quicker remyelination of damaged tissue after ischemic stroke.

Our results are supported by a plethora of studies showing the beneficial effects of BMSC transplantation on white matter after ischemic stroke. Mice and rats subjected to middle cerebral artery occlusion (MCAO) displayed an increased expression of MBP or O4 following BMSC transplantation compared to the MCAO group devoid of BMSC treatment [32,33]. Similarly, exposure to the secretome of endothelial progenitor cells enhanced OPC differentiation, myelin production and behavioral outcomes in mice subjected to prolonged hypoperfusion [34]. These studies support our finding that OPCs produced more myelin when treated with BMSCs, which may be due to the variety of growth factors secreted by these stem cells cultured in an environment containing ischemic cells [35]. In particular, brain-derived neurotrophic factor (BDNF) has been extensively identified as a major modulator of oligodendrocyte lineage function [36,37,38]. Indeed, BDNF positive T_regs_ are upregulated compared to other BDNF positive CD4^+^ T-cells in the human brain following ischemic stroke, suggesting that T_regs_ may influence oligodendrocyte differentiation via the BDNF pathway [39]. Moreover, BDNF/TrkB activity is postulated to activate sonic hedgehog (Shh) signaling, which mediates oligodendrocyte differentiation [40,41]. A-kinase anchor protein has also been implicated recently as an important modulator of OPC differentiation, implicating that certain proteins may interact with T_regs_ in directing OPC-mediated myelination [42]. While the results of our study support previous research about white matter injury and the therapeutic benefits of stem cells, they also provide evidence that T_regs_ act as a key mediator of myelination, perhaps through the BDNF pathway.

Although our previous study showed that T_regs_ are powerful negative regulators of pro-inflammatory cytokine secretion, specifically IL-6, the results of the current study reveal a unique neuroprotective property of IL-6 in OPCs [14]. We saw an increase in IL-6 levels in the BMSC co-culture group containing T_regs_. This may be due to increased BMSC activity in this group, which is supported by our results that there was increased FGF-β secretion in both the BMSC T_reg^+^_ and BMSC no magnet group. Although many studies have labeled IL-6 as a pro-inflammatory cytokine, there is equally compelling evidence suggesting that it also has neuroprotective properties and that it is involved in crosstalk with the BDNF pathway [43,44]. Indeed, the treatment of ischemic pheochromocytoma cells with mesenchymal stem cells produces therapeutic effects due to IL-6 and vascular endothelial growth factor secretion [45]. IL-6 has also been shown to confer protection against excitotoxic cell death in neurons [46]. Therefore, IL-6 seems to have dual functions in the brain and more research is warranted to elucidate the effects of this cytokine on ischemic OPCs specifically.

As stated above, the group treated with BMSCs containing T_regs_ displayed increased MBP production compared to the T_reg^−^_ depleted BMSCs. To date, limited studies demonstrate a potential interaction between T_regs_ and BDNF/Shh signaling in the brain. Of note, Shh signaling was necessary for expansion of T_regs_ by dendritic cells, but this was not in the context of the CNS [47]. This information would aid in elucidating the mechanism by which T_regs_ result in increased MBP expression. Additionally, treatment of cells belonging to the oligodendrocyte lineage showed an increase in myelination after exposure to IL-6 fused with its receptor in a dose-dependent manner [48]. This further supports the notion that IL-6 may have beneficial effects in the CNS and could serve as a positive regulator of OPC differentiation. Moreover, bone marrow-derived-mast cell production of IL-6 was enhanced after exposure to cell-cell contact with T_regs_ [49]. More specifically, T_regs_ use the cytokine transforming growth factor beta to modulate levels of IL-6 secreted by mast cells, controlling the inflammatory milieu and increasing neutrophil clearance with increasing levels of IL-6 secretion [50]. Therefore, IL-6 may have anti-inflammatory properties which influence differentiation of OPCs, and T_regs_ may be direct modulators of its secretion. Effective treatments targeting white matter injury will ensure survival, differentiation and protection of OPCs from inflammation, as this is critical for the remyelination process [51]. More specifically, the complex interactions between BDNF, IL-6, T_regs_ and BMSCs should be investigated to further elucidate the unique roles that each plays in influencing the oligodendrocyte lineage.

Recognizing that BMSCs are unlikely to transdifferentiate into myelin-producing oligodendrocytes before six days after induction, we did not include stem cell markers in our immunocytochemistry assays in order to distinguish between the original OPCs and stem cells [52]. Future studies investigating the communication between OPCs and BMSCs should include this to better differentiate between the cells and obtain a more definitive cell count. Staining for an earlier OPC lineage marker, such as NG2, in combination with intermediate and later markers would similarly allow characterization of OPCs at all levels of maturity. In this same vein, we were unable to explore the differences in surviving cell number between the groups treated with BMSCs due to their direct introduction into the cell culture. Cell therapy with T_regs_ or stem cells has shown to result in increased cell viability in neurons and oligodendrocytes after ischemic conditions [14,29,30]. We found that BMSC treatment resulted in healthier oligodendrocytes, but the effects on OPC cell number may be addressed in future experiments. In addition, this study was conducted completely in vitro, and therefore has constraints regarding clinical application. Future studies should focus on examining the in vivo effects of T_regs_ on white matter injury due to ischemic stroke to segue into the clinic. 

In conclusion, we found that BMSC co-culture was associated with increased differentiation of OPCs after ischemic injury and that T_reg_ exposure resulted in more robust expression of MBP. These results confirmed prior research highlighting the beneficial effects of BMSCs after ischemic stroke. Our novel finding consisted of the increased expression of MBP due to exposure to T_regs_, but the exact mechanism by which this occurs is unclear at this stage. Limitations including the restricted clinical application and the lack of visualization of stem cell markers should be taken into consideration in future research. However, the finding that T_regs_ resulted in overall healthier oligodendrocytes emphasizes their potential therapeutic benefit to white matter recovery after ischemic stroke.

## Figures and Tables

**Figure 1 jcm-08-00537-f001:**
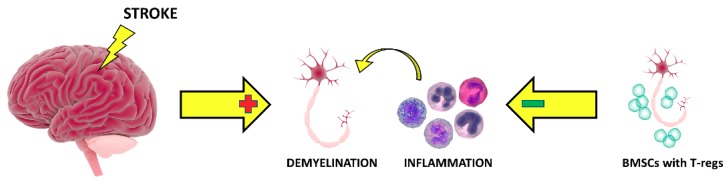
T-regulatory cells sequester stroke-induced inflammation leading to less demyelination.

**Figure 2 jcm-08-00537-f002:**
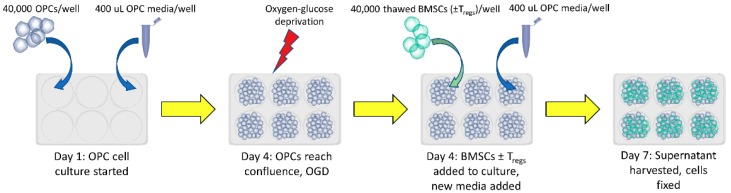
Oligodendrocyte progenitor cells (OPCs) were cultured for 3 days and then subjected to oxygen-glucose deprivation and reoxygenation (OGD/R). Bone marrow-derived stem cells (BMSCs) with or without T_regs_ were then added directly into the cell culture, and both cell types were cultured together for 3 more days.

**Figure 3 jcm-08-00537-f003:**
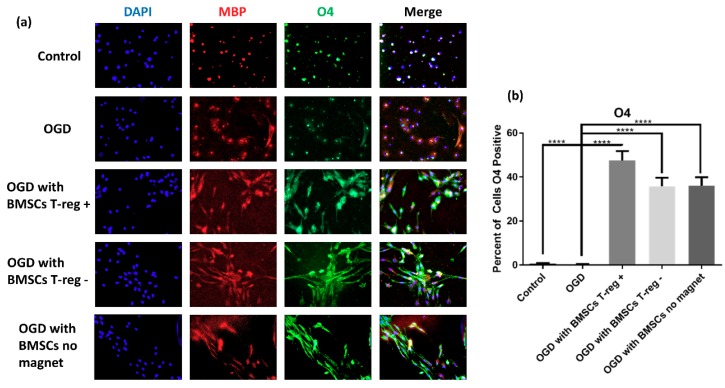
OPCs co-cultured with BMSCs showed increased expression of intermediate oligodendrocyte marker O4 compared to control. (**a**) Fluorescently labeled cell nuclei with DAPI, oligodendrocyte progenitor cell (OPC) myelin processes labeled with myelin basic protein, OPC O4 expression labeled with oligo O4, and merge. (**b**) Quantification of MBP+ cell count in each group shown as percent of cells expressing O4 per image, mean ± SEM.

**Figure 4 jcm-08-00537-f004:**
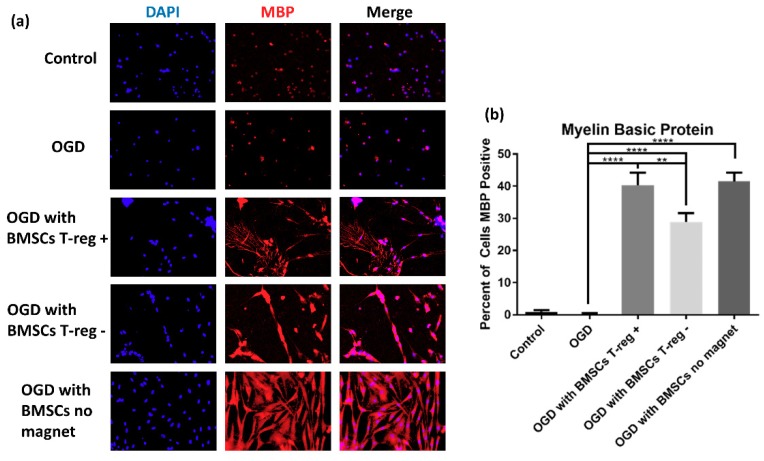
BMSCs with T_regs_ confer increased myelination compared to BMSCs without T_regs_. (**a**) Fluorescently labeled cell nuclei with DAPI, OPC myelin processes labeled with myelin basic protein (MBP), and merge. (**b**) Quantification of MBP+ cell count in each group shown as percent of cells expressing MBP per image, mean ± SEM.

**Figure 5 jcm-08-00537-f005:**
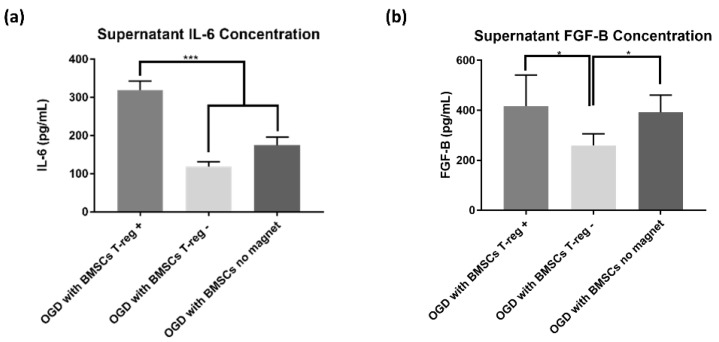
BMSCs containing T_regs_ displayed increased IL-6 secretion. (**a**) Concentration of interleukin 6 (IL-6) in cell supernatant shown as mean ± SD. (**b**) Concentration of fibroblast growth factor beta (FGF-β) in cell supernatant shown as mean ± SD.

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
