# Peer review of "T-Regulatory Cells Confer Increased Myelination and Stem Cell Activity after Stroke-Induced White Matter Injury"

_jcm, 2019, doi:10.3390/jcm8040537_

Round 1
Reviewer 1 Report
In this study, the authors investigated the role of Treg in BMSCs on the OPC survival and differentiation in OGD model. The results indicated that OPCs cultured with BMSCs containing Tregs show greater cell viability, less pro-inflammatory cytokine secretion, and greater myelin production after exposure to OGD. The current study reveals a unique finding that IL-6 which is considered as pro-inflammatory factor, is neuroprotective against OGD in OPCs and that Tregs may be direct modulators of IL-6 secretion. However, some key data do not support the conclusions 1. Whether Treg or BMSCs rescues OPC from OGD is critical, but the data are missing. 2. There is a good overlay of MBP with O4 based on immunostaining in control and OGD group in Figure 3A. However, the quantitative analysis showed no MBP+ cells in O4+ cells in control and OGD group in Figure 3B. 3. Based on Figure 4A immunostaining, almost all DAPI staining are overlaid with MBP in BMSC Treg- group, which contradicts with the quantification data Figure 4b . 4. To dissect the role of BMSC-released cytokines on oligodendrocytes differentiation/survival, the authors focused on IL6 and FGF-β, why not BDNF which has been extensively identified as a major modulator of oligodendrocyte lineage function? 5. In most of the results, there is no difference between BMSC Treg+ and BMSCs no magnet groups, such as the O4 counts, MBP counts and FGF-β. Does this support the use of Treg alone rather than BMSC for cel therapy in white matter injury? 6. The term BMSC Treg+, BMSC Treg- and BMSCs no magnet groups are very confusing. Do they refer to Treg+, BMSC/Treg_ and BMSC?
Author Response
Reviewer #1:
In this study, the authors investigated the role of Treg in BMSCs on the OPC survival and differentiation in OGD model. The results indicated that OPCs cultured with BMSCs containing Tregs show greater cell viability, less pro-inflammatory cytokine secretion, and greater myelin production after exposure to OGD. The current study reveals a unique finding that IL-6 which is considered as pro-inflammatory factor, is neuroprotective against OGD in OPCs and that Tregs may be direct modulators of IL-6 secretion.
Response: We thank the Reviewer for taking the time to review our manuscript and for highlighting our novel finding.
1. Whether Treg or BMSCs rescues OPC from OGD is critical, but the data are missing.
Response: We agree that the way that BMSCs and/or T-regs affect cell viability after OGD is of significance and we thank the Reviewer for raising this important point. Since we directly inoculated the culture with BMSCs before the cell count, it is not possible to accurately delineate the exact numbers of OPCs left in their respective groups after the cell culture. There is evidence that both BMSCs and T-regs help cell survival after ischemia in other published works, and we have added a short section to our discussion to address this point and clarify this important concept in the field.
2. There is a good overlay of MBP with O4 based on immunostaining in control and OGD group in Figure 3A. However, the quantitative analysis showed no MBP+ cells in O4+ cells in control and OGD group in Figure 3B.
Response: In Figure 3, many of the cells in the control and OGD groups contain positively stained MBP and O4 nuclei as the Reviewer mentioned. However, a stained nucleus only does not indicate typical morphology/staining of the oligodendrocyte lineage. For the purpose of quantitative analysis, we only counted the cells as MBP/O4 positive if the cell displayed processes extending from the nucleus, as can be seen with many of the BMSC groups. For example, in the OGD Merge picture from Figure 3, there is one cell off to the right that shows processes staining for both MBP and O4 extending north and south of the nucleus. This cell was considered positive for the purposes of our analysis, but nuclei staining only was not evident. We did state this in section 2.4 of the methods section, and will reiterate this in the results section to clarify for the reader.
3. Based on Figure 4A immunostaining, almost all DAPI staining are overlaid with MBP in BMSC Treg- group, which contradicts with the quantification data Figure 4b .
Response: We thank the Reviewer for pointing this out and seek to clarify this in our manuscript. As mentioned in the previous response, cells must have displayed stained processes extending out from the nucleus to be considered positive for the purposes of our quantitative analysis, and we strictly adhered to this rigorous criterion in our analysis of all groups. We hope that our added section in the results section will clarify any discrepancies seen with stained nuclei and the quantitative analysis.
4. To dissect the role of BMSC-released cytokines on oligodendrocytes differentiation/survival, the authors focused on IL6 and FGF-β, why not BDNF which has been extensively identified as a major modulator of oligodendrocyte lineage function?
Response: We thank the Reviewer for the comment about BDNF. We agree that BDNF is extensively present in regulation pathways pertaining to the oligodendrocyte lineage. Our previous study (Neal et. al. 2018) focused on FGF-β and IL-6, so we saw our study as a continuation of the investigation into T-regs and those particular cytokines. In addition to the comments we previously made about BDNF in the discussion, we have now added a comment identifying BDNF specifically as an appealing target for future investigation.
5. In most of the results, there is no difference between BMSC Treg+ and BMSCs no magnet groups, such as the O4 counts, MBP counts and FGF-β. Does this support the use of Treg alone rather than BMSC for cel therapy in white matter injury?
Response: We thank the Reviewer for pointing this out and understand that the particular treatments that each group underwent may be confusing. The only difference between these two groups is that the cells in the BMSCs no magnet group were run through the depletion column prior to inoculation. Therefore, the data to which the Reviewer refers suggest that running the cells through the depletion column did not affect our results. We cannot conclude that T-regs alone are sufficient for cell therapy. In the results, we clarify the difference between these two groups to make this less confusing for the reader.
6. The term BMSC Treg+, BMSC Treg- and BMSCs no magnet groups are very confusing. Do they refer to Treg+, BMSC/Treg_ and BMSC?
Response: As mentioned above, the BMSC no magnet group consisted of BMSCs with their native T-regs. These cells were run through the depletion column to more closely mimic the conditions of the T-reg- BMSCs. Since the groups had very similar results, we can conclude that the depletion column did not significantly affect the outcome of the study. We have now added a section about this in the discussion to minimize further confusion.
Reviewer 2 Report
Title: T regulatory cells confer increased myelination and stem cell activity after stroke induced white matter injury.
This study describes the beneficial effect of Bone marrow-derived stem cells with depleted T-regulatory cells on oligodendrocytes progenitor cells after ischemia.
This paper can be published with some minor changes. There are questions that need to be answered.
In the methods section, particularly in the figure 2 showing the experimental design, it is not clear whether the BMSC alone or including the T-regulatory cells are added to the culture. Considering the results, this is important to clarify if the BMSC alone, T regulatory cells alone or both are responsible for the beneficial effect seen on oligodendrocytes progenitor cells, in this study.
If we compare the figure 3 and 4, particularly if we consider the control and the OGD groups, the DAPI positive cells in OGD are less in term of number in figure 4, compared to the controls. While the DAPI positive cells in OGD and control groups are quite similar in figure 3. How this could be explained?
In the result section, the graphs show that there is more O4 (figure 3b) and MBP (figure 4b) positive cells respectively, in OGD with BMSCs T-reg+ (Positive) compared to OGD with BMSCs T-reg- (Negative) group. However, the two groups (BMSCs T-reg+ versus OGD with BMSCs T-reg-) in term of immunostaining O4 and MBP in (figure 3a) and (figure 4a), respectively are quite similar. For the staining, were the ROI (regions of interest) observed, sufficiently representative?
Author Response
Reviewer #2:
This study describes the beneficial effect of Bone marrow-derived stem cells with depleted T-regulatory cells on oligodendrocytes progenitor cells after ischemia.
This paper can be published with some minor changes. There are questions that need to be answered.
Response: We thank the Reviewer for highlighting the major findings of our study.
1. In the methods section, particularly in the figure 2 showing the experimental design, it is not clear whether the BMSC alone or including the T-regulatory cells are added to the culture. Considering the results, this is important to clarify if the BMSC alone, T regulatory cells alone or both are responsible for the beneficial effect seen on oligodendrocytes progenitor cells, in this study.
Response: We thank the Reviewer for bringing our attention to this. Both BMSCs and T-regulatory cells were added to the culture. In view of these, we believe that both T-regs and BMSCs are responsible for the results. We have now clarified this in Figure 2 and in the methods section.
2. If we compare the figure 3 and 4, particularly if we consider the control and the OGD groups, the DAPI positive cells in OGD are less in term of number in figure 4, compared to the controls. While the DAPI positive cells in OGD and control groups are quite similar in figure 3. How this could be explained?
Response: We agree that in Figure 3, the number of DAPI stained nuclei in the OGD group are similar to other groups. We chose the OGD photo in Figure 3 to highlight the particular cell on the right of the picture, which developed myelin processes similar to cells in the BMSC groups. We hoped that this would serve as a comparison of the myelination difference between the OGD only group and the BMSC groups. However, the vast majority of the photos more closely resemble the OGD group in Figure 4, approximately equating to 56% of the cell viability of the control group. In our previously study with neurons (Neal et. al. 2018), we found that OGD decreased the number of viable cells to a similar percentage of the control. We have now clarified this in the results section.
3. In the result section, the graphs show that there is more O4 (figure 3b) and MBP (figure 4b) positive cells respectively, in OGD with BMSCs T-reg+ (Positive) compared to OGD with BMSCs T-reg- (Negative) group. However, the two groups (BMSCs T-reg+ versus OGD with BMSCs T-reg-) in term of immunostaining O4 and MBP in (figure 3a) and (figure 4a), respectively are quite similar. For the staining, were the ROI (regions of interest) observed, sufficiently representative?
Response: We thank the Reviewer for the comments about the immunocytochemistry. As mentioned above, we only counted the cells as positive if they displayed processes beyond the nucleus, and therefore the staining can look similar between the two groups at first glance. In Figure 3, there was no significant difference between the BMSC groups. In Figure 4, there was about a 10% difference in MBP positive cells between the BMSC T-reg+ group and BMSC T-reg- group. In order to systematically analyze the ICC, we obtained 32 pictures from each plate, with 4 photos per well (1 each quadrant). We kept this constant throughout the entire analysis, and thus we believe that the data used for the quantitative analysis are representative sampling of the treatment conditions. We have added a comment to the methods further clarifying the way in which we analyzed the plates.
Round 2
Reviewer 1 Report
The authors responded well and revised the manuscript. No more comments